# Genotypic Variability of Photosynthetic Parameters in Maize Ear-Leaves at Different Cadmium Levels in Soil



**Mario Franić [1,2] , Vlatko Galić [3,* ] , Zdenko Lončarić [4] and Domagoj Šimić [2,3]**

1   Institute of Agriculture and Tourism, Karla Huguesa 8, 52 440 Poreč, Croatia; mario@iptpo.hr
2   Centre of Excellence for Biodiversity and Molecular Plant Breeding, Svetošimunska Cesta 25, 10 000 Zagreb, Croatia; domagoj.simic@poljinos.hr
3   Agricultural Institute Osijek, Južno Predgrađe 17, 31 000 Osijek, Croatia
4   Faculty of Agrobiotechnical Sciences Osijek, Josip Juraj Strossmayer University of Osijek, Vladimira Preloga 1, 31 000 Osijek, Croatia; zdenko.loncaric@fazos.hr
*   Correspondence: vlatko.galic@poljinos.hr; Tel.: +385-31515525

**Abstract:** Excess of cadmium (Cd) in soil leads to a number of adverse effects which challenge agricultural production. Aims of this study were to investigate variations in Cd content in leaves (*Zea mays* L.) of different maize genotypes and to detect effects of Cd on photosynthesis through chlorophyll *a* fluorescence. Pot vegetation experiments with four maize genotypes and four different soil Cd levels were repeated for two years. Chlorophyll fluorescence, photosynthetic pigments and inductively coupled plasma (ICP) analysis for ear-leaf Cd and zinc (Zn) and soil Cd were carried out. Significant differences between genotypes were found for leaf Cd, where higher Cd soil concentrations resulted in higher Cd leaf concentrations. Cd uptake into maize leaves increased with increasing Cd levels in soil, which was genotype-dependent and higher and lower Cd accumulating groups were formed. Changes in chlorophyll fluorescence caused by elevated Cd levels in soil were mostly visible as changes in dissipation energy, yields of primary photosystem II photochemistry and electron transport. Decrease of reaction centers per antenna chlorophyll and increased variable fluorescence at J step ($V_J$) resulted in decrease of performance indexes in the highest Cd concentration. Decreases in chlorophyll fluorescence parameters suggest reduced functionality of reaction centers and problems in re-oxidation of primary quinone acceptor ($Q_A$).

**Keywords:** cadmium; chlorophyll fluorescence; photosynthesis; trace metal stress; inbred lines

## 1. Introduction

Toxic trace metals are a global problem in agriculture affecting not only crop plants but indirectly also animals and humans. Due to human activities such as mining, smelting, application of commercial fertilizers and sewage sludge, metal pollution is becoming a risk to many ecosystems [1,2]. Cadmium (Cd) is a non-essential metal with no known physiological function. Exposure of plants to Cd leads to alterations in many cellular processes and functions such as photosynthetic activity, antioxidant activity, ion channels, plant water status and redox imbalance [3–6] as well as to reduction of cell proliferation and growth [7,8].

Cd, as a non-essential metal, causes many adverse effects in plant functionality. Plants vary in their ability for Cd uptake, as well as in thresholds for its toxicity, but generally, 5–10 μg Cd/g plant dry mass is expected to induce toxicity in most higher plants [9]. A widely used threshold for Cd in soil extracts is 1 mg/kg [10], while the concentrations in European soils vary widely from 0.02 to 3.17 mg/kg with mean of 0.09 mg/kg [11]. Contamination of soil by a single metal is rare and usually where one metal is concentrated there are others present in higher concentrations as well. Metals also tend to

interact, and these interactions combined with the effects of soil on their uptake and toxicity [12] are hard to predict. It is generally assumed that toxic trace metals enter plant cells through transporters of essential metals and trace metal uptake is in competition with uptake of essential metals, such as K, Ca, Mg, Fe, Mn [9,13,14]. However, mechanisms of Cd uptake and translocation have not yet been properly elucidated. Cd accumulation is, besides pH [15], largely affected by genotype (Cd excluders and non-excluders), variation in acidification of root rhizosphere between species, soil temperature, level of evapotranspiration and also by chemical and physical parameters of the soil in the experiment which differs from soil in the field. The primary mechanism for Cd intracellular immobilization is through formation of Cd-phytochelatin complexes; Cd is deposited in vacuoles and Cd ions are translocated by xylem and phloem, where translocation is genotype specific.

The cause of Cd toxicity could originate from its similarity to zinc (Zn) which is an essential metal to biological systems and an important factor in the protection of biological membranes against oxidative stress [16], but is also regarded as toxic when present in higher concentrations [17]. Toxicity is thus manifested by competing with essential divalent cations for protein binding sites causing imbalances in metal homeostasis [18].

A large number of studies have been performed on the effect of Cd on photosynthetic machinery—from isolated thylakoid membranes to hydroponically grown plants but relatively few studies have focused on the influence of Cd on the photosynthetic apparatus in plants grown from seeds on soil polluted with Cd [19]. Plotting chlorophyll fluorescence values during the first second upon illumination of dark-adapted photosynthetic tissue with actinic light on a logarithmic scale, from minimum $F_0$ (O) to maximum $F_m$ (P) reveals a polyphasic rise, with two intermediate steps J and I [20]. A procedure for biophysical interpretation of this fluorescence transient (O-J-I-P), known as JIP-test, was developed by Strasser and Strasser [21]. O-J-I-P transients and JIP-test parameters were shown to be reliable indicators of stress [22–28]. The JIP-test is being used in investigating stress physiology in a number of plant species under controlled and field conditions [29]. JIP-test gives information about changes in photochemistry efficiency and heat dissipation and is widely used for assessment of plant reaction to various types of stress conditions [30–34]. Negative effects of Cd on photosynthesis, especially on photosystem II (PSII), are easily detectable using methods which measure chlorophyll *a* fluorescence [35–37].

Uptake of Cd in maize inbred lines varies [38], and this variability appears to be caused by different alleles of only a single gene [39] coding for Zn/Cd transporting ATPase, possibly from Heavy Metal ATPases (HMA) family [40]. In accordance with variability in uptake, Cd differently affects photosynthetic machinery in different maize inbreds [41]. Higher doses of Cd are expected to induce more severe physiological responses [3], but some cultivars are able to alleviate these effects [42]. Synergistic effects of Cd/Zn toxicity can also be observed [43], as additive interaction of their uptake is managed through competition for transporter sites. Cd/Zn synergistic toxicity can disrupt uptake of many essential metals and cause deterioration of photosynthetic machinery [44]. Some studies demonstrated that synergistic uptake of Cd and Zn could be reduced by impairing the HMA4 homologs [45].

Major aims of this study were to investigate (1) the variation of Cd uptake in maize leaves of four different maize genotypes, (2) detect the effects of Cd uptake on photosystem of selected genotypes through chlorophyll fluorescence, and to (3) identify if there is possible tolerance or sensitivity of the selected genotypes to Cd.

## 2. Materials and Methods

### 2.1. Soil Preparation

Soil was collected from the growing field of Agricultural Institute Osijek, from the top 30 cm layer and air-dried with mixing for 30 days in a greenhouse. A quantity of 1400 kg of air-dried soil was

taken and sieved through a 5 mm sieve. Soil was divided into four equal parts, 350 kg of soil per each part. Soil chemical properties are presented in Table 1.

**Table 1.** Soil properties (mean ± SE, $n = 2$ for organic matter (%), $P_2O_5$-AL, $K_2O$-AL, $CaCO_3$ (%); $n = 8$ for all other parameters).

| Element | Value | Element | Value |
|---|---|---|---|
| pHKCl | 6.99 ± 0.03 | Cu (mg kg$^{-1}$) | 24.00 ± 0.19 |
| pHH$_2$O | 8.05 ± 0.02 | Fe (mg kg$^{-1}$) | 29358.75 ± 208.80 |
| Organic matter (%) | 2.57 ± 0.08 | Mn (mg kg$^{-1}$) | 676.05 ± 5.53 |
| $P_2O_5$-AL (%) | 29.58 ± 0.64 | Zn (mg kg$^{-1}$) | 69.40 ± 0.64 |
| $K_2O$-AL (%) | 25.60 ± 3.43 | Ni (mg kg$^{-1}$) | 31.32 ± 0.22 |
| $CaCO_3$ (%) | 1.26 ± 0 | Co (mg kg$^{-1}$) | 13.83 ± 0.11 |
| Pb (mg/kg) | 20.50 ± 0.20 | Cr (mg kg$^{-1}$) | 44.87 ± 0.56 |

According to World Resource Base (WRB) classification [46], soil type was determined as eutric gleysol with 1.5% sand, 71.2% silt, and 27.2% clay. Soil was neutral to weak alkaline reaction, slightly calcareous, and had moderate fertility with medium humus content (Table 1). Soil potassium was in the range of moderate availability (class C) and phosphorus in range high availability (class D) determined by AL-acetic acid method [47]. The total concentrations of all trace metals determined after extraction by *aqua regia* were very low, mainly lower than 50% of maximum allowed concentrations (MAC) in agricultural soils, and less than 25% of MAC for Pb (20.5 mg kg$^{-1}$) and Cd (0.1 mg kg$^{-1}$).

Soil was contaminated with $CdCl_2$ (Sigma-Aldrich, St. Louis, USA) solution to three different Cd levels: 0.5, 1 and 5 mg of Cd per kilogram of soil (mg Cd kg$^{-1}$ soil) while control soil was left uncontaminated. For treatment preparation, soil was spread in a few centimeters thick layer and sprayed (using a spray bottle) with prepared $CdCl_2$ solution. Plastic 12 L pots (r = 275 mm, h = 250 mm) with drainage holes were filled with prepared soil, 14 kg of air-dry soil per pot, 4 pots per genotype each representing a single biological replicate ($n_{biol}$ = 4), 16 pots per each treatment giving total of 64 pots for the experiment. Cd concentrations in soil per treatment were determined by inductively coupled plasma optical emission spectroscopy (ICP-OES) before the experiment. Control treatment, Cd0.5, Cd1 and Cd5 treatments had (mean ± SE): 0.11 ± 0.01, 0.62 ± 0.04, 1.07 ± 0.46 and 4.89 ± 0.06 mg Cd kg$^{-1}$ soil, respectively.

### 2.2. Plant Material and Growth Conditions

Seeds of four maize (*Zea mays* L.) genotypes (B73, Mo17, B84, Os6-2) were planted in pots in the beginning of May (9 May 2012; 7 May 2013). Soil in pots was completely soaked with water and left to drain for three days prior to planting, to reach field water capacity. Planting was done by hand, seeds were planted 5 cm deep in four replications with eight seeds per pot. Pots were initially watered with 200 mL of water daily and the amount of water was increased up to 2 L as the plants grew. Watering was adjusted according to Penman–Monteith equation for evapotranspiration [48]. At six-leaf stage, plants were thinned to four plants per pot. Fertilization was performed according to the recommendations based on the soil analysis. Plants were grown to physiological maturity (R6 stage) in a greenhouse until V3 phase after which pots were transferred outside to the field environmental conditions. Experiment was set as completely randomized and pots were shuffled randomly every morning along with watering. There were differences in two growing years mostly in temperature and precipitation. 2012 had a little less rainfall than 2013 but both years were in the normal rainfall range in July when measurements were made. 2012 was warmer than 2013; deviations from the normal average temperature were 0.4, 3.0 and 3.6 °C for May, June and July 2012, respectively. Deviations from the normal average temperature in 2013 were 0.2, 0.5 and 1.8 °C for May, June and July, respectively. Cumulative insolation duration was higher in 2012 and it was above the average normal cumulative insolation duration.

### 2.3. Chlorophyll Fluorescence Measurements

Chlorophyll *a* fluorescence was measured in the first half of July (2012 and 2013) on attached leaves, during flowering (tasseling) when maize plants are particularly susceptible to stress, using Plant Efficiency Analyser (Handy PEA, Hansatech, King's Lynn, UK). Measurements for both years were recorded early in the morning (7–9 a.m.) due to midday depression of photosynthesis in maize [49]. Temperatures during measurements ranged between 19 and 21 °C. Chlorophyll *a* fluorescence was measured on the middle section of the upper side of ear-leaves on four plants per pot (technical replicates), making 16 measurements per genotype (four pots with four plants each, *n* = 16) for each genotype-treatment combination. Ear leaves were chosen as they were shown to best represent the overall plant chlorophyll content [50]. After dark adaptation time of 30 minutes' chlorophyll fluorescence transient was induced by applying a pulse of saturating red light (peak at 650 nm, 3200 mmol m$^{-2}$ s$^{-1}$) provided by 3 ultra-bright LED's. LED´s are focused via lenses on the leaf surface that is exposed by the leaf clip (4 mm in diameter). Saturating light pulse induces chlorophyll a fluorescence increase from minimal fluorescence ($F_0$), when all reaction centers open, to maximal fluorescence ($F_m$), when all reaction centers closed. A total of 120 data points were collected during the 1 s measurement. Chlorophyll *a* fluorescence data were processed with PEA plus software version 1.10, provided with the Plant Efficiency Analyser. Data points and parameters obtained by chlorophyll *a* fluorescence were analyzed according to the JIP-test that outputs multiple parameters quantifying the photochemistry of PSII. JIP-test was described by Strasser et al. [51–54]. Parameters are given in Table S1.

### 2.4. Photosynthetic Pigments and Dry Weight Analysis

Photosynthetic pigments were analyzed in 2013 only. Leaf tissue was ground in a porcelain mortar in liquid nitrogen with addition of magnesium hydroxide carbonate (Sigma-Aldrich). One gram of sample was transferred to an Eppendorf tube with 1 milliliter of extra pure cold acetone (Fisher chemical) and vortexed. Tubes were put on ice for 15 min, centrifuged for 10 min at 4 °C and 14,000 rpm. Precipitate was re-extracted with the same procedure until the tissue lost its color. Concentration of photosynthetic pigments was determined from absorbance peaks at 470, 661.6 nm and 664.8 nm measured by spectrophotometer (Specord 200, Analytik Jena, Jena, Germany) in a glass cuvette. Acetone was used as blank. Pigment concentrations were determined according to Lichtenthaler [55] by the following formulae: Chlorophyll $a = 11.24A_{661.6} - 2.04A_{664.8}$; Chlorophyll $b = 20.13A_{644.8} - 4.19A_{661.6}$; Chlorophyll $a + b = 7.05A_{661.6} + 18.09A_{644.8}$; Total carotenoids $= (1000A_{470} - 1.90C_a - 64.14C_b)/214$.

Percentage of dry weight was determined by weighing 1 g of ground fresh sample (ground in liquid nitrogen) in an Eppendorf tube and drying at 105 °C for 48 h. Dry weight is expressed as % of fresh weight. Pigments and dry weights were measured in 3 repetitions for each genotype in each treatment.

### 2.5. ICP-OES Analysis—Plant Material and Soil

Ear-leaves were collected after chlorophyll *a* fluorescence measurements. Samples from each pot were put in separate paper bags and dried. After drying samples were milled with a heavy metal free mill (ZM 200, Retsch, Haan, Germany). The leaf samples were digested with 10 mL of a 5:1 mixture of HNO$_3$ (Fisher chemical, Leicestershire, UK) and H$_2$O$_2$ (J.T.Baker, Deventer, The Netherlands) at 180 °C for 60 min in microwave oven (Mars 6, CEM, Matthews, USA). After cooling, total Cd concentrations were measured using ICP-OES (Optima 2100 DV, PerkinElmer, Wellesley, USA). Leaf samples were analyzed with an internal pooled plasma control and with the reference material (Rice flour, IRMM-804, Sample No. 0533, European Commission, Joint Research Centre, Institute for Reference Materials and Measurements, Geel, Belgium) prepared in the same way as the other leaf samples.

The total trace metal concentrations in soil samples were analyzed after grinding using heavy metal free grinder (ZM 200, Retsch, Haan, Germany), sieving through the sieves of 2 mm, and digesting

with 10 mL of a 3:1 mixture of HCl (Fisher chemical) and $HNO_3$ (ISO 11466) at 210 °C for 60 min in microwave oven (CEM Mars 6, Matthews, NC, USA). The total Fe, Mn, Zn, Cu, Ni, Co, Cr, Pb and Cd concentrations in digested soil samples were measured by ICP-OES (Optima 2100 DV, PerkinElmer, Wellesley, USA). Soil samples were analyzed with an internal pooled plasma control and with the reference material (Loam soil, ERM CC141, Sample No. 0037, European Commission, Joint Research Centre, Institute for Reference Materials and Measurements, Geel, Belgium) prepared in the same way as were the soil samples extracted by aqua regia.

## 2.6. Statistical Analysis

Statistical analysis was performed using R [56]. For three-way factorial analysis of variance, Anova function from "car" package [57] was used with four main sources of variation: treatment, genotype, year and replication as well as treatment–genotype, treatment–year, genotype–year and genotype–treatment-year interactions. Package "agricolae" [58] was used for LSD tests and "Hmisc" [59] package was used to calculate Pearson´s correlation coefficients and their respective *p*-values.

## 3. Results

Definitions and formulas for all analyzed chlorophyll *a* fluorescence parameters are given in Table S1. Analysis of variance for all selected chlorophyll *a* fluorescence parameters, Cd and Zn concentrations in the leaf has revealed significant effects (at $p < 0.05$ level or lower) of main sources of variation (genotype, treatment, year), as well as their interactions. Only exceptions were $ET_0/RC$, $DI_0/RC$, N, $RC/CS_0$ and $t_{max}$ for factor "Year", Cd for corresponding interactions including "Year" and $TR_0/RC$, $TR_0/ABS$, N, $RC/CS_0$, $ET_0/(ET_0\text{-}TR_0)$ for "Treatment × Year" interaction (Table S2). Analysis of variance for photosynthetic pigments and dry weight has revealed highly significant effects of treatment: genotype interaction for all parameters was significant ($p < 0.01$), except Chl *a*/Chl *b*/DW where the difference between genotypes was not significant.

## 3.1. ICP Analysis for Cd and Zn

ICP analysis of Cd in ear-leaves shown on Figure 1A revealed that Cd accumulation is different in four analyzed genotypes. Mo17 and Os6-2 lines accumulate large amounts of Cd that are concentration-dependent while B73 and B84 accumulate very little Cd in leaves.

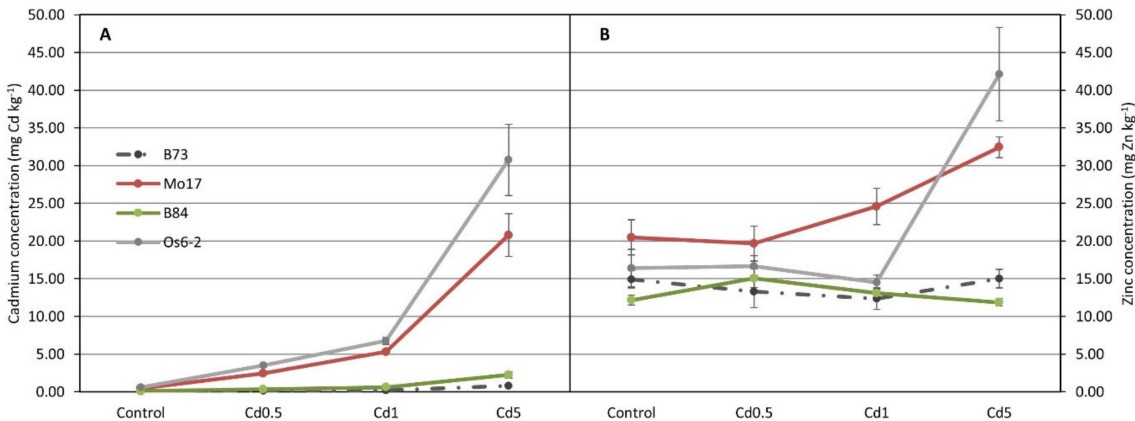

**Figure 1.** Concentrations of cadmium (Cd) (**A**) and zinc (Zn) (**B**) in four maize genotypes (B73, Mo17, B84 and Os6-2) grown on four different levels of Cd (control, 0.5, 1 and 5 mg Cd per kg of soil; Cd applied as $CdCl_2$) measured on ear-leaf during flowering by inductively coupled plasma optical emission spectroscopy (ICP-OES) (mean ± SE, *n* = 6). Genotypic effects for both Cd and Zn accumulation were significant at $\alpha = 0.001$ (Table S2).

Maize line Os6-2 accumulated highest amounts of Cd in all treatments, even in control. Highest amount of Cd accumulated by that line was 29.95 and 31.55 mg kg$^{-1}$ (on dry weight basis) in 2012 and in 2013, respectively. Smallest amount of Cd in leaves was accumulated by B73 line with values of 1.01 and 0.54 mg kg$^{-1}$ (on dry weight basis) in 2012 and in 2013, respectively. The graph clearly separates maize lines into two groups—ones that accumulate high levels of Cd in leaves (non-excluders) and the ones that accumulate very small amounts of Cd in leaves (excluders, Figure 1A). Accumulation of Zn showed similar behavior as Cd accumulation and it can be seen from Figure 1B that, as in Cd accumulation, it is genotype-dependent but the separation of genotypes is not as clear as in Cd accumulation.

## 3.2. Chlorophyll a Fluorescence Transients

Polyphasic chlorophyll *a* fluorescence transients measured on ear-leaves of investigated genotypes showed characteristic OJIP curve shape when plotted on a logarithmic scale, for both control and Cd treated plants (Figure 2). Transients were plotted as kinetics of the relative variable fluorescence (Vt) [53].

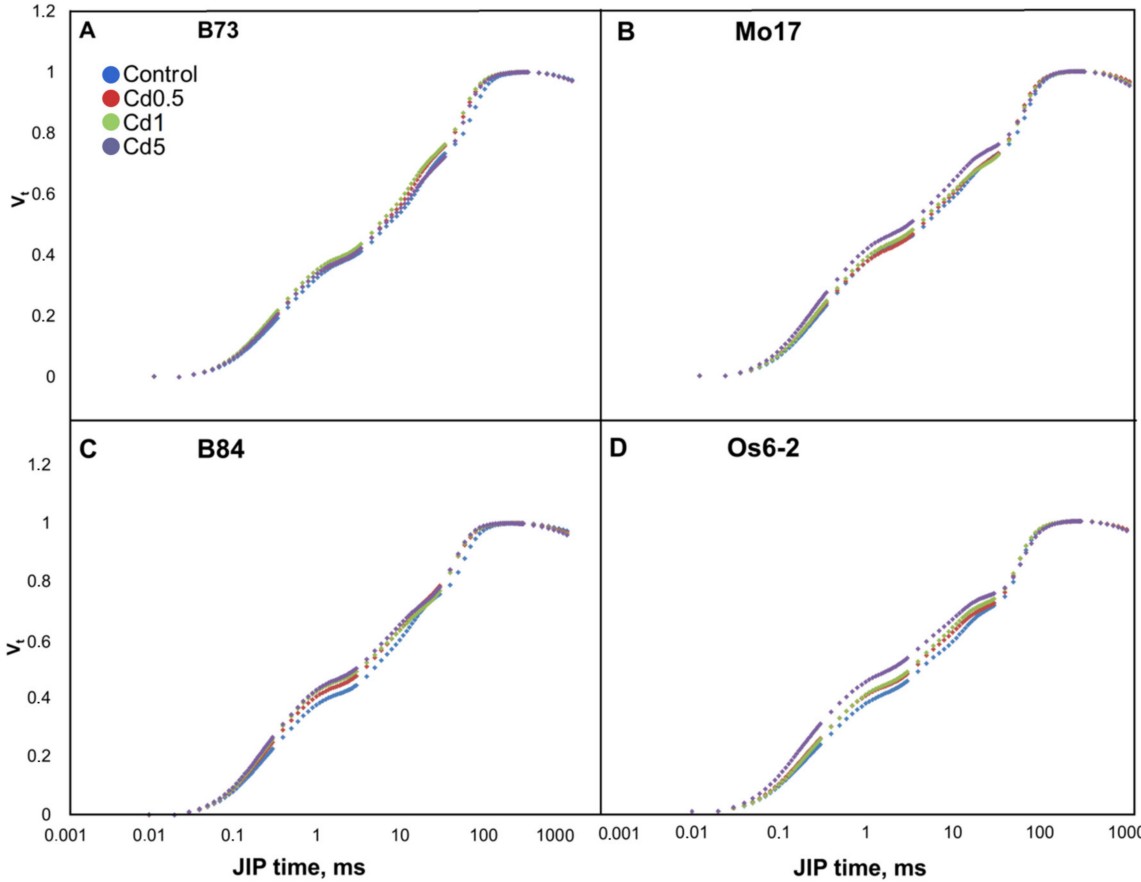

**Figure 2.** Normalized chlorophyll *a* fluorescence transients from 0 ms (O) through J, I and P of four maize genotypes ((**A**) B73, (**B**) Mo17, (**C**) B84 and (**D**) Os6-2) grown on four different levels of Cd in soil (control, 0.5, 1 and 5 mg Cd per kg of soil; Cd applied as CdCl$_2$) measured on ear-leaves during tasseling. Transients represent averages from 2012 (*n* = 16) and 2013 (*n* = 12).

Shape of the fluorescence transients in B73 line did not differ from the control except for a very small rise in I step in Cd0.5 and 2. In Mo17 most noticeable difference in the transients was in Cd5 where J and I steps increased. B84 line showed an increase in J step in all treatments and Os6-2 line

showed an increase in J and I steps that was most evident in Cd5. Changes in fluorescence transients are visible in the photochemical phase (O-J) and in thermal phase (J-I-P) (for terminology see [60]).

### 3.3. JIP-Test Parameters

JIP-test parameters (for formulae and definitions see Supplementary Table S1) were normalized to control plants and the deviations of Cd stress treatments from the control parameter values are shown on radar plots for each genotype (Figure 3). It can be seen from Figure 3 that genotypes reacted differently to elevated levels of Cd in soil. In general, B73 line appears to be least affected by increased Cd levels in soil (seen as smallest decreases in $PI_{ABS}$ and $PI_{total}$) and there is practically no difference in yields of three fluxes: $TR_0/ABS$, $ET_0/ABS$ and $ET_0/TR_0$.

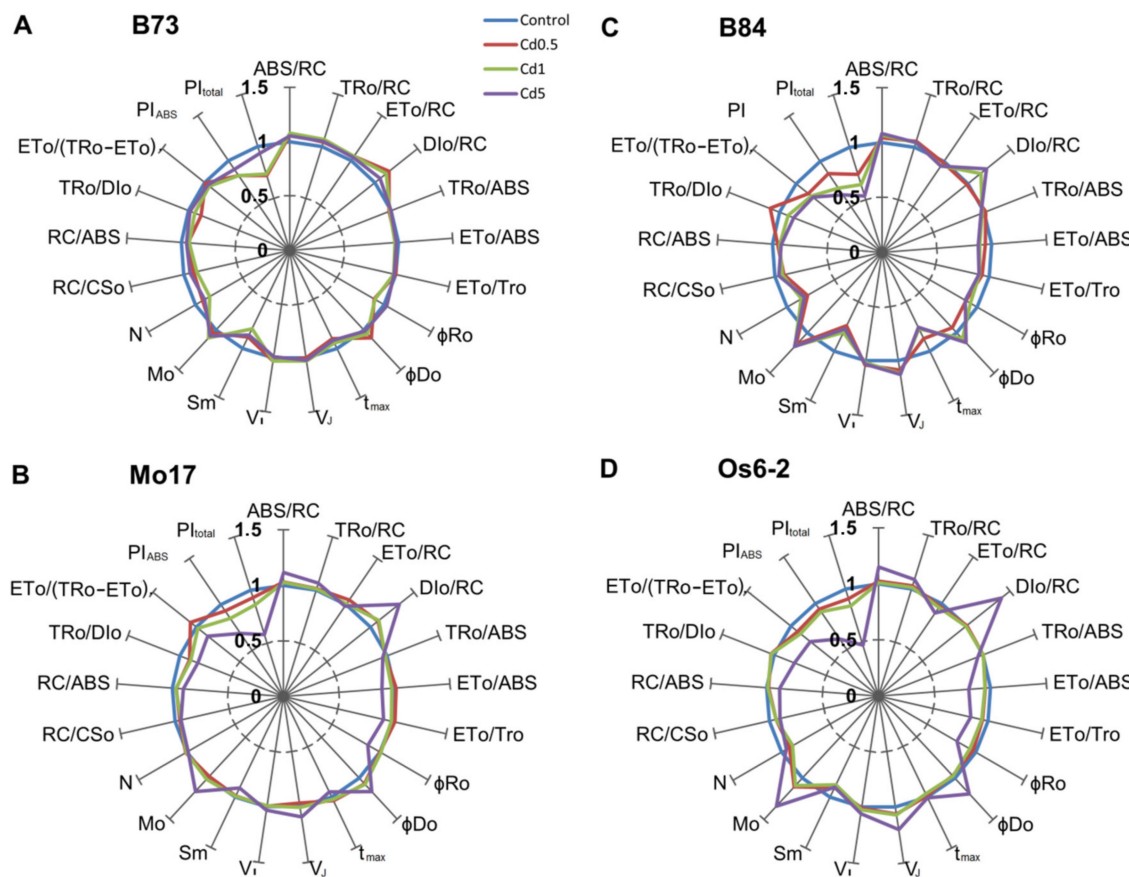

**Figure 3.** Effects of four different levels of Cd in soil (control, 0.5, 1 and 5 mg Cd per kg of soil; Cd applied as $CdCl_2$) on selected functional and structural JIP-test parameters in four maize genotypes ((**A**) B73, (**B**) Mo17, (**C**) B84 and (**D**) Os6-2) plotted to their respective controls (radar plot center = 0, maximum = 1.5, control value = 1.0). Values represent averages from 2012 (*n* = 16) and 2013 (*n* = 12). For definitions and formulas see Table S1. Effects of treatments were significant at $\alpha$ = 0.01 for all analyzed JIP-test parameters (for full results of ANOVA, see Table S2).

Additionally, very small differences are visible in relative variable fluorescence at J and I steps ($V_J$ and $V_I$). B73 line appeared to be less affected by the highest Cd level in soil (Cd5) and more negatively affected by the two intermediate levels which is especially noticeable in $PI_{ABS}$, $PI_{total}$ and $DI_0/RC$ parameters. On the other hand, OS6-2 line showed completely different pattern of reactions. Two intermediate Cd levels had little effect on biophysical parameters of Os6-2 line, while the highest Cd level caused large changes in almost all parameters. Lines B84 and Mo17 showed similar patterns of reactions with the exception of Cd0.5 and Cd1 did not affect B84 line as much as Mo17.

Maximum quantum yield of primary photochemistry ($TR_0/ABS$) in B73 was decreased more in Cd0.5 and Cd1 than in Cd5 (0.980, 0.989 and 0.998; values normalized to control) while in other three genotypes decrease was noticeable in Cd1 and more greatly in Cd5. A slight decrease is noticeable in Cd0.5 in line Mo17 in $TR_0/ABS$ suggesting the decreased efficiency of PSII. Changes in electron transport activity can be observed using parameters associated with electron transport: $ET_0/ABS$, $ET_0/TR_0$, $\varphi_{Ro}$, $\delta_{Ro}$. Reduction in $ET_0/ABS$ and $ET_0/TR_0$ implies that the probability for the electron transport beyond $Q_A^-$ is decreased and that the maximum yield of electron transport beyond $Q_A$ decreased. From the radar plots it can be seen that electron transport beyond $Q_A$ was least affected in B73 line which shows almost no deviation from the control in any of the Cd treatments. Other three genotypes show a decrease in mentioned parameters, most prominent in Cd1 and Cd5. $\varphi_{Ro}$ and $\delta_{Ro}$ represent quantum yield and efficiency/probability for reduction of end electron acceptors at the PSI side. Reduction of these parameters was most noticeable in Cd5 treatment, except in B73 where the first two treatments had the greatest effect and in B84 line where all treatments similarly reduced $\varphi_{Ro}$ and $\delta_{Ro}$ parameters.

Increased ABS/RC, average absorption per active reaction center ($Q_A$ reducing), suggests that certain degree of RCs is inactivated or that the apparent antenna size increased. ABS/RC values increased in all four genotypes but in Mo17 and Os6-2 increase was largest in Cd5 and in other two genotypes ABS/RC increased similarly across all treatments. This increase in ABS/RC is associated with the decrease in active RCs per excited cross section ($RC/CS_0$) and RC density on a chlorophyll basis (RC/ABS). Increase in trapping per RC ($TR_0/RC$) under Cd treatments, which can be seen in all genotypes in Cd5 treatment, can indicate impairment of the oxygen evolving complex. Specific flux for electron transport ($ET_0/RC$) increased in B73 and Mo17 (most noticeable on Cd0.5) and decreased in B84 and Os6-2. Most evident change in the specific energy fluxes per RC was in energy dissipation ($DI_0/RC$) and it increased in all genotypes and for all treatments except Cd0.5 in B84 and treatments 1 and 2 in Os6-2 line.

Two indexes of plant performance, performance index for energy conservation form exciton to the reduction system of intersystem electron acceptors ($PI_{ABS}$) and performance index for energy conservation form exciton to the reduction of PSI acceptors ($PI_{total}$) showed decreases that followed the rise of Cd concentration in soil, with Cd5 causing the largest decrease (except in B73 line where the lowest PI indexes were in Cd0.5 and Cd1). $PI_{ABS}$ is used to quantify PSII behavior and depends on three components: RC/ABS, $TR_0/DI_0$ and $ET_0/(TR_0-ET_0)$. $PI_{total}$ depends on several components: $TR_0/ABS$, $ET_0/TR_0$, $Chl_{RC}/Chl_{total}$ and $\varphi_{Ro}$.

Plotting log $(PI_{ABS})_{rel}$ against $(ET_0/ABS)_{rel}$ is considered a typical property of plants to transform absorbed light energy into chemical energy that is diverted to further metabolic reactions. The log function of relative performance index on absorption basis (log $(PI_{ABS})_{rel}$) was linearly correlated with the probability that an absorbed photon moves $Q_A$-$(ET_0/ABS)_{rel}$ ($R^2 = 0.53$). All genotypes in the control treatment had positive values while increasing Cd concentration in soil caused a decrease in all genotypes; this suggests a decrease in efficiency of transforming light energy into chemical energy (NADPH) with increased Cd concentration in soil. Particularly low values were for Os6-2, B84 and Mo17 in Cd5 treatment (all in the negative quadrant) while B73 was least affected by Cd treatments and its values were not below zero (Figure 4).

B84 line performed poorly also in Cd1 treatment which was also in the negative quadrant. From Figure 1B, high performing (B73) and low performing (B84, Mo17, Os6-2) genotypes can be observed.

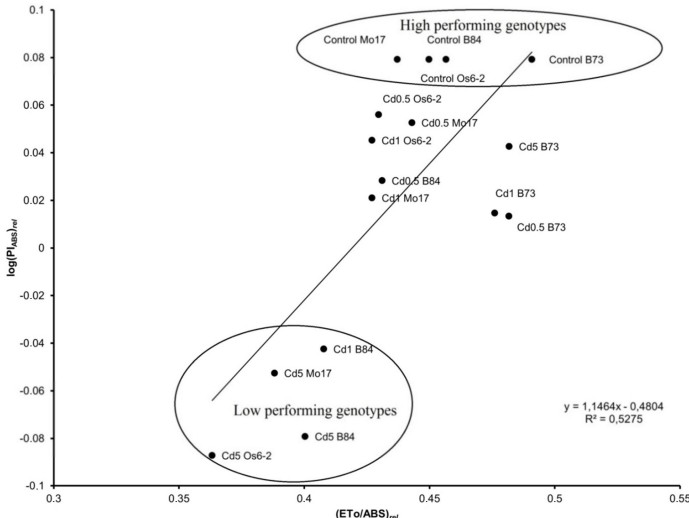

**Figure 4.** Linear correlation between the log function of relative performance index on absorption basis (log(PIABS)rel) relative yield of electron transport (ET0/ABS) in four maize genotypes (B73, Mo17, B84 and Os6-2) grown on four different levels of Cd (control, 0.5, 1 and 5 mg Cd per kg of soil; Cd applied as $CdCl_2$). Values represent averages of 2012 and 2013 ($n = 16$, $n = 12$, respectively). For definitions and formulas see Table S2.

## 3.4. Photosynthetic Pigments and Dry Weight

Spectrophotometric measurement data for chlorophylls and carotenoids revealed genotype-dependent changes caused by increased Cd concentration in soil. In inbred line B73, dry weight (DW) decreased with increasing Cd levels in Cd0.5 and Cd1 and increased in Cd5. Other three genotypes showed more or less concentration-dependent increase in dry weight (Table 2).

**Table 2.** Dry weight, chlorophyll and carotenoid content expressed by dry weight of four maize genotypes (B73, Mo17, B84 and Os6-2) grown at four different levels of Cd in soil (control, 0.5, 1 and 5 mg Cd per kg of soil; Cd applied as $CdCl_2$) determined in ear-leaves during tasseling in 2013 (Mean ± SE, $n = 3$). Means with different letters are significantly different ($p < 0.05$).

| | Cd Excluders | | Cd Non-Excluders | |
|---|---|---|---|---|
| | **B73** | **B84** | **Mo17** | **Os6-2** |
| Dry weight (%) | | | | |
| Control | 25.25 ± 0.06 [a] | 23.38 ± 0.04 [c] | 20.35 ± 0.49 [c] | 21.95 ± 0.07 [b] |
| Cd0.5 | 22.80 ± 0.19 [c] | 22.98 ± 0.18 [d] | 22.89 ± 0.21 [b] | 21.20 ± 0.27 [c] |
| Cd1 | 20.95 ± 0.08 [d] | 25.84 ± 0.01 [a] | 23.26 ± 0.03 [b] | 22.40 ± 0.08 [b] |
| Cd5 | 23.90 ± 0.08 [b] | 25.32 ± 0.15 [b] | 24.14 ± 0.05 [a] | 24.90 ± 0.10 [a] |
| $LSD_{0.05}$ | 0.3742 | 0.8805 | 0.3894 | 0.4971 |
| Chl *a*/DW | | | | |
| Control | 5.01 ± 0.13 [b] | 4.66 ± 0.87 [c] | 5.71 ± 0.21 [a] | 4.85 ± 0.03 [a] |
| Cd0.5 | 5.27 ± 0.07 [b] | 6.04 ± 0.06 [a] | 4.40 ± 0.07 [b] | 4.90 ± 0.04 [a] |
| Cd1 | 5.88 ± 0.07 [a] | 5.46 ± 0.16 [b] | 4.35 ± 0.03 [b] | 4.13 ± 0.04 [b] |
| Cd5 | 4.33 ± 0.07 [c] | 4.53 ± 0.06 [c] | 3.52 ± 0.05 [c] | 3.74 ± 0.02 [c] |
| $LSD_{0.05}$ | 0.2912 | 0.3769 | 0.3680 | 0.1172 |

**Table 2.** *Cont.*

| | Cd Excluders | | Cd Non-Excluders | |
|---|---|---|---|---|
| | **B73** | **B84** | **Mo17** | **Os6-2** |
| **Chl *b*/DW** | | | | |
| Control | 1.40 ± 0.05 [b] | 1.36 ± 0.32 [b] | 1.56 ± 0.01 [a] | 1.30 ± 0.06 [a] |
| Cd0.5 | 1.34 ± 0.02 [b] | 1.75 ± 0.02 [a] | 1.18 ± 0.03 [b] | 1.32 ± 0.06 [a] |
| Cd1 | 1.63 ± 0.01 [a] | 1.66 ± 0.06 [a] | 1.15 ± 0.00 [b] | 1.10 ± 0.04 [b] |
| Cd5 | 1.09 ± 0.02 [c] | 1.22 ± 0.04 [b] | 0.95 ± 0.02 [c] | 0.96 ± 0.03 [c] |
| LSD$_{0.05}$ | 0.0961 | 0.0568 | 0.1480 | 0.0616 |
| **Chl *a* + *b*/DW** | | | | |
| Control | 6.41 ± 0.08 [b] | 6.01 ± 1.18 [c] | 7.27 ± 0.21 [a] | 6.15 ± 0.06 [a] |
| Cd0.5 | 6.61 ± 0.09 [b] | 7.79 ± 0.05 [a] | 5.57 ± 0.09 [b] | 6.22 ± 0.06 [a] |
| Cd1 | 7.51 ± 0.06 [a] | 7.13 ± 0.09 [b] | 5.50 ± 0.02 [b] | 5.24 ± 0.04 [b] |
| Cd5 | 5.42 ± 0.09 [c] | 5.76 ± 0.11 [c] | 4.47 ± 0.07 [c] | 4.70 ± 0.03 [c] |
| LSD$_{0.05}$ | 0.2590 | 0.3927 | 0.2830 | 0.1603 |
| **Car/DW** | | | | |
| Control | 1.32 ± 0.03 [c] | 1.27 ± 0.27 [b] | 1.35 ± 0.04 [a] | 1.25 ± 0.02 [a] |
| Cd0.5 | 1.40 ± 0.01 [b] | 1.59 ± 0.01 [a] | 1.09 ± 0.01 [c] | 1.23 ± 0.02 [a] |
| Cd1 | 1.70 ± 0.02 [a] | 1.54 ±0.04 [a] | 1.23 ± 0.01 [b] | 1.25 ± 0.01 [a] |
| Cd5 | 1.42 ± 0.02 [b] | 1.30 ± 0.02 [b] | 0.99 ± 0.01 [d] | 1.16 ± 0.00 [b] |
| LSD$_{0.05}$ | 0.0641 | 0.0726 | 0.0820 | 0.0485 |
| **Chl *a*/Chl *b*/DW** | | | | |
| Control | 14.25 ± 0.86 [b] | 14.75 ± 0.46 [ab] | 18.03 ± 1.00 [a] | 16.96 ± 0.24 [ab] |
| Cd0.5 | 17.26 ± 0.16 [a] | 15.03 ± 0.40 [a] | 16.33 ± 0.31 [ab] | 17.50 ± 0.12 [a] |
| Cd1 | 17.23 ± 0.32 [a] | 12.76 ± 0.85 [b] | 16.24 ± 0.17 [b] | 16.73 ± 0.31 [b] |
| Cd5 | 16.64 ± 0.18 [a] | 14.68 ± 0.37 [b] | 15.27 ± 0.19 [b] | 15.60 ± 0.22 [c] |
| LSD$_{0.05}$ | 1.556 | 1.763 | 1.652 | 0.7586 |
| **Chl *a* + *b*/Car/DW** | | | | |
| Control | 19.27 ± 0.13 [b] | 20.23 ± 0.78 [b] | 26.39 ± 0.65 [a] | 22.49 ± 0.17 [b] |
| Cd0.5 | 20.71 ± 0.29 [a] | 21.33 ± 0.20 [a] | 22.39 ± 0.27 [b] | 23.91 ± 0.17 [a] |
| Cd1 | 21.11 ± 0.09 [a] | 17.87 ± 0.20 [c] | 19.30 ± 0.05 [c] | 18.66 ± 0.04 [c] |
| Cd5 | 15.98 ± 0.08 [c] | 17.51 ± 0.06 [c] | 18.74 ± 0.05 [c] | 16.27 ± 0.14 [d] |
| LSD$_{0.05}$ | 0.5611 | 1.149 | 0.5218 | 0.4589 |

All other measured chlorophyll- and carotenoid-related parameters in Mo17 and Os6-2 lines decreased gradually as Cd concentration in soil increased. B73 and B84 showed more complex responses to increased Cd concentrations in soil. In B73 line increasing Cd concentrations in soil caused an increase in most of the chlorophyll and carotenoid related parameters (except in Chl *b*/DW) but in Cd5 treatment those parameters decreased. In B84 line slightly different pattern can be seen, smallest Cd concentration (Cd0.5) caused the largest increase in chlorophyll and carotenoid parameters values and as Cd concentration increased those values gradually decrease. The exception is Chl *a*/Chl *b*/DW where Cd1 treatment caused a decrease while Cd5 treatment caused an increase Chl *a*/Chl *b*/DW parameter (Table 2). Mo17 and Os6-2 genotypes show a similar gradual decreasing pattern with increased Cd levels (exception is DW where values increase with increasing Cd levels) while other two genotypes (B73, B84) show a totally different and more noisy pattern (Table 2). This grouping of genotypes is similar to grouping by Cd and Zn accumulation where B73 and B84 form one low-accumulating group and Mo17 and Os6-2 form high-accumulating group (Figure 1A,B).

### 3.5. Correlation Analysis between JIP-Test Parameters and Trace-Metal Concentrations

Os6-2 and Mo17 (non-excluders) accumulated larger amounts of Zn than B73 and B84 (excluders). In general, Cd and Zn accumulation in ear-leaves had strong positive correlation ($r = 0.65$, $p < 0.001$) (Table 3). Interestingly, Zn accumulation had higher correlation ($r = 0.57$, not shown) with Cd accumulation in lines that accumulate large amounts of Cd (Os6-2, Mo17) while in lines that accumulate small amounts of Cd (B73, B84) correlation was lower ($r = 0.42$, not shown).

**Table 3.** Correlations between Cd and Zn concentrations (leaf and soil) determined by inductively coupled plasma optical emission spectroscopy (ICP-OES) analysis and chlorophyll *a* fluorescence parameters of four maize genotypes grown at four different levels of Cd in soil (control, 0.5, 1 and 5 mg Cd per kg of soil; Cd applied as $CdCl_2$) determined in ear-leaves during tasseling ($n = 32$). *, **, *** indicate significant correlations at $p < 0.05$, $p < 0.01$ and $p < 0.001$ level, respectively, while n.s. indicates non-significant correlation. For definitions and formulas of parameters see Table S1.

| Variable | $V_J$ | $M_0$ | TRo/ABS | ETo/ABS | Eto/Tro |
|---|---|---|---|---|---|
| Zn (mg kg$^{-1}$) | 0.67 ** | 0.74 *** | −0.65 ** | −0.67 ** | −0.67 ** |
| Cd (mg kg$^{-1}$) | 0.73 *** | 0.82 *** | −0.76 *** | −0.74 *** | −0.73 *** |
| Soil Cd (mg kg$^{-1}$) | 0.51 * | 0.55 * | −0.54 * | −0.52 * | −0.51 * |
| **Variable** | **RC/CSo** | **RC/ABS** | **TRo/DIo** | **ETo/(TRo-ETo)** | **PI$_{abs}$** |
| Zn (mg kg$^{-1}$) | −0.63 ** | −0.67 ** | −0.54 * | −0.54 * | −0.52 * |
| Cd (mg kg$^{-1}$) | −0.58 * | −0.79 *** | −0.65 ** | −0.59 * | −0.60 * |
| Soil Cd (mg kg$^{-1}$) | −0.20 n.s. | −0.58 * | −0.44 n.s. | −0.45 n.s. | −0.44 n.s. |
| **Variable** | **ABS/RC** | **TRo/RC** | **ETo/RC** | **DIo/RC** | **DIo/CSo** |
| Zn (mg kg$^{-1}$) | 0.75 *** | 0.71 ** | −0.60 * | 0.73 ** | 0.48 n.s. |
| Cd (mg kg$^{-1}$) | 0.86 *** | 0.81 *** | −0.62 ** | 0.84 *** | 0.64 ** |
| Soil Cd (mg kg$^{-1}$) | 0.59 * | 0.56 * | −0.40 n.s. | 0.59 * | 0.56 * |
| **Variable** | **PI$_{total}$** | **φDo** | **Zn (mg kg$^{-1}$)** | **Cd (mg kg$^{-1}$)** | **Soil Cd (mg kg$^{-1}$)** |
| Zn (mg kg$^{-1}$) | −0.34 n.s. | 0.62 n.s. | | 0.65 *** | 0.34 n.s. |
| Cd (mg kg$^{-1}$) | −0.46 * | 0.73 ** | 0.65 *** | | 0.63 ** |
| Soil Cd (mg kg$^{-1}$) | −0.40 n.s. | 0.53 n.s. | 0.34 n.s. | 0.63 ** | |

Significant correlations were found between chlorophyll *a* fluorescence parameters and parameters of ICP-OES analysis (soil Cd, leaf Cd and Zn). Quantum yields ($TR_0/ABS$, $ET_0/ABS$, $ET_0/TR_0$) were negatively correlated with both soil and leaf Cd concentration, likewise electron transport ($ET_0/RC$) and electron transport beyond $Q_A^-$ ($ET_0/(TR_0 - ET_0)$) were also negatively correlated with leaf Cd concentration. Both performance indexes ($PI_{ABS}$, $PI_{total}$) were negatively correlated with Cd leaf concentration (Table 3).

Negative correlations were also observed between density of reaction centers on chlorophyll a basis (RC/ABS) and leaf Cd and between density of reaction centers per excited cross section ($RC/CS_0$) and both leaf and soil Cd. Positive correlations were observed between soil and leaf Cd and parameters that suggest impaired functionality of PSII: $DI_0/RC$, $DI_0/CS_0$, $φD_0$, $V_J$, $M_0$, and consequently ABS/RC, $TR_0/RC$ (Table 3).

## 4. Discussion

### 4.1. Effects of Cd on O-J-I-P Transients

Toxic trace metals impair the process of photosynthesis at several steps; consequently it is reasonable to assume that plants under trace metal stress will be challenged by oxidative stress, which has been shown previously [9,61,62]. Indeed, increased levels of Cd in soil caused genotype-dependent changes in concentration and photosynthetic machinery detectable by changes in chlorophyll *a* fluorescence transients, JIP-test parameter values, dry mass and photosynthetic pigments.

Chlorophyll a fluorescence transients can be separated into two groups: no visible change between treatments (B73) and visible changes in chlorophyll fluorescence transients (B84, Mo17, Os6-2) (Figure 2). Differences from control transients were mostly visible in J and I steps and showed gradation with increasing Cd concentrations more clearly than in B73. Related to ICP-OES analysis, Cd non-excluders (Mo17, Os6-2, Figure 1A) show Cd-induced changes in chlorophyll fluorescence transients while in Cd excluders group (Figure 1A) B73 does not show changes and B84 shows changes similar to Cd non-excluder group even though it accumulated very small amounts of Cd. JI phase of the transient represents changes in PQ-pool reduction [63]. Increase in J step, also visible as increase in VJ value, reflects the start of QA re-oxidation by secondary electron acceptor (QB) [21] so an increase in J step would suggest a problem in QA re-oxidation and a consequent build-up of reduced QA. Direct effect of Cd on the electron transfer between QA and QB could be explained through interaction with non-heme iron involved in this step [64].

*4.2. Effects of Cd on JIP-Test Parameters*

Increase in I step, quantified as variable fluorescence at I step ($V_I$), suggests accumulation of reduced plastoquinone which is unable to transfer electrons to dark reactions [65]. Reduced normalized area above the OJIP curve ($S_m$) represents diminished ability to transport electrons per reaction center, thus suggesting that less energy is needed to close all reaction centers [52]. $S_m$ was especially reduced in B73 and B84 lines, less reduced in Os6-2 line and reduced only in Cd5 in Mo17 line. $S_m$ is proportional to the number of electrons that pass through the electron transport chain [60,66] and the parameter N, derived from $S_m$, is the number of times $Q_A$ becomes reduced and re-oxidized until maximum fluorescence (Fm) is reached. N decreases with the increase of Cd concentration in soil and along with the decrease in $S_m$ suggests that maximum fluorescence would be reached quicker because less electrons is needed to reduce PSII electron acceptors which can be seen through decrease of $t_{max}$ parameter. Photosynthetic yield of PSII ($TR_0/ABS$) was not affected by Cd treatments in B73 line (Figure 3) while other lines showed decrease in quantum yield of PSII most notably seen in the highest Cd concentration (Cd5). Increases in $TR_0/RC$ which are most apparent in treatments with the highest Cd concentration in soil can indicate impairment of oxygen evolving complex [65]. Changes in photosynthetic yield are reflected through decreases in quantum yield of electron transport ($ET_0/ABS$) and the efficiency of trapped exciton to move an electron into the electron transport chain further than primary quinone acceptor ($ET_0/TR_0$). This suggests that Cd causes photoinhibitory damage to PSII which has been suggested in previous studies [67]. In Cd non-excluder genotypes (Mo17, Os6-2) Cd0.5 and Cd1 treatments had little effect while Cd5 treatment caused negative changes in basically all presented JIP-test parameters (Figure 3) suggesting that the functioning of the photosystem was more compromised with more accumulated Cd. These genotypes seem to accumulate Cd (probably along with Zn) and have no mechanism of coping with excess Cd. In Cd excluder genotypes (B73, B84) radar plots look very different. B73 showed very little change in all treatments and even increase of $PI_{ABS}$ and its respective components in Cd5 compared to Cd05 and Cd1 treatments. In B73, the favorable allele of Cd/Zn transporting ATPase is expressed, alleviating the negative effects of Cd accumulation in maize leaves [39]. Higher dose of Cd in soil possibly triggered the expression of this ATPase, partially mitigating the negative effects of Cd treatment. B84 was negatively affected by all treatments, especially Cd1 and Cd5, even though it accumulated very little Cd and its radar plot looks more similar to Cd non-excluder radar plots (Figure 2) indicating Cd sensitivity in B84 line. Interestingly, in one research [41] B84 and Os6-2 maize inbred lines at seedling stage were used to test the effects of Cd on photosynthetic parameters and the study showed B84 as sensitive and Os6-2 as Cd insensitive. This suggests that different strategies could be implemented to cope with excess Cd at different stages of growth. This is also in concordance with research on other plant species [68,69].

### 4.3. Performance of Light Utilization in Cd Stressed Plants

The relationship between log $PI_{ABS}$ and $ET_0/ABS$ (or the driving force for photosynthesis of the observed system and probability that an absorbed photon moves $Q_A^-$) is considered a typical property of a plant to transform light into chemical energy (NADPH) and further directed to biochemical processes of photosynthesis; hence the linear correlation between these two parameters reflects the decrease in light energy transformation ability under elevated Cd concentration in soil (Figure 1B). This relationship has been previously used to evaluate performance of plant species in stressful conditions [28,31,70]. By plotting log function of $PI_{ABS}$ against $ET_0/ABS$ genotypes separated into high (B73) and low (Mo17, B84, Os6-2) performing at different Cd levels in soil.

### 4.4. Photosynthetic Pigments

Spectrophotometric measurement and characterization of chlorophylls and carotenoids revealed changes that are genotype- and Cd-dependent. Chlorophylls, carotenoids and dry weight data are shown on Table 3. A concentration-dependent decrease in chlorophyll *a* and *b* in Mo17 and Os6-2 is evident (38% and 24%, respectively) and 25% decrease in chlorophyll content can be taken as a clear indication of chlorophyll damage by generation of $^1O_2$ [71]. In Mo17 and Os6-2 decreases in chlorophyll *a* and *b* are followed by a decrease in Chl *a/b* ratio. Significant decrease in chlorophyll content under Cd stress has been previously reported by Shaw [72] in common bean (*Phaseolus vulgaris*) and Sandalio et al. [73] in pea (*Pisum sativum*) plants. Chlorophyll *a* and chlorophyll *b* weight ratio (Chl a/b) is an indicator of the functional pigment pool and light adaptation of the photosynthetic apparatus [74]. Chlorophyll *a* is found in reaction centers of PSI and PSII and in the pigment antenna, while chlorophyll *b* is found exclusively in the pigment antenna system. Decrease in Chl a/b ratios can be interpreted as an increase in antenna system of PSII [75,76]. Decrease in Chl *a/b* ratio indicates preferential degradation of Chl *a* over Chl *b*; Chl *a* is present in larger quantities and is in close association with the reaction centers and hence it is more exposed to $^1O_2$ attack than Chl *b* which constitutes the outer antenna of PSII and PSI [77]. Chlorophyll *a* and *b* weights and their relationships in Mo17 and Os6-2 lines are simpler than in B73 and B84 lines. While in Mo17 and Os6-2, Cd causes a concentration-dependent decrease of Chl *a*, *b* and their ratio in B73 and B84 Cd, in certain concentrations, it has the opposite effect. For example, in both B73 and B84, Cd5 caused a significant increase in chlorophyll *a* and *b* contents. In B84 line, increase in chlorophylls is accompanied by a decrease in Chl *a/b* and Chl/Car ratios indicating oxidative damage to chlorophyll *a* [78] while in B73 line these two parameters increased. Increases in chlorophyll content could be attributed to an adaptive process in the plant to keep up the photosynthetic activity like that of the control [79] or to inhibition of tissue elongation according to Maksymiec and Baszyński [78]. In this case, increase in Chl *a/b* ratios in B73 line was due to increase in the amount of Chl *a*. Weight ratios of Chl *a* and *b* to total carotenoids is an indicator of greenness of plants and a decrease in this parameter is an indicator of stress and damage to the photosynthetic apparatus and this parameter was more affected by Cd in Mo17 and Os6-2 lines, especially in Cd5.

Dry weight of Mo17 and Os6-2 genotypes was affected similarly by increased levels of Cd and is seen as a Cd concentration-dependent increase in dry weight. The same is visible in B84 with a little less intensity but not in B73 where dry weight decreased slightly and Cd1 (and not Cd5) seemed to have greatest effect on the decrease. Decrease of plant dry weight at higher Cd concentrations has been extensively reported [7,80,81]. This decrease could probably be attributed to negative effects on cell cycle and inhibition of cell division as reported by Hendrix et al. [8] in *Arabidopsis thaliana*. Increase in dry weight in Mo17 and Os6-2 could be attributed to higher Zn accumulation in elevated levels of Cd in soil. As reported by Köleli et al. [82] decreases in dry weight from increasing Cd concentrations are more severe in Zn deficient plants; Zn protects plants from Cd toxicity by improving plants defense against oxidative stress. It seems that the effect of elevated Cd levels in soil produces a response, in maize genotypes that were used in this research, which is heterotic group related. Heterotic groups represent the concept of dividing genetic material originating from similar historical sources that

exhibit heterosis when crosses are performed among individuals with contrasting background and are used extensively in seed production [83]. Cd non-excluders Os6-2 and Mo17 both originating from Lancaster pool showed similar responses which are simple-pigment values decrease with increasing Cd levels in soil, while Cd excluders B73 and B84 belonging to Stiff stalk pool show different and more complex responses.

*4.5. ICP-OES Analysis*

Studies on accumulation of Cd and Zn reveal mostly antagonistic interaction [84,85], but synergistic interactions were reported as well [86–88]. As shown by [89] synergistic effects could be observed in loamy soil where Zn uptake increases with applied Cd. McKenna et al. [90] reported that Cd stimulated the uptake of Zn in young leaves of lettuce (*Lactuca sativa* L.). Furthermore, McKenna et al. [91] reported in a research study on Alpine Penny-cress (*Thlaspi caerulescens*) grown in hydroponic culture an increase in Zn accumulation with Cd treatments with a significant strong positive correlation. They also suggested that the difference in Cd and Zn accumulation between two investigated ecotypes is caused by more than one gene, although in maize Cd accumulation is probably controlled by only a single gene as suggested by [38], and that accumulation and tolerance are genetically independent traits. Furthermore, different alleles of a single gene coding for Cd/Zn transporting ATPase were found between inbred lines B73 and Mo17 [39]. Similarly, in our investigation there is a synergistic relationship between Cd and Zn uptake (Figure 1, Figure 4) and the connection between uptake of these two metals can be seen from their correlation which is strong, positive and significant ($r = 0.65$, $p < 0.001$). Possible mechanism involved in this synergistic transport is gene HMA4 coding for *Heavy Metal Associated* ATPase [40] congruent with findings of [39]. Correlations between soil Cd and plant Cd were strong and positive ($r = 0.63$, $p < 0.01$), which were expected and confirmed already by established results in this type of experiments. Florijn et al. [92] classified maize plants as Cd excluders and Cd non-excluders, by that classification B73 and B84 would be Cd excluders and Mo17 and Os6-2 would be Cd non-excluders. In their research Cd excluder was B73 line (Stiff stalk heterotic group) and Cd non-excluder was H98 line (Lancaster heterotic group). In our research ICP-OES analysis of Cd accumulation in leaves revealed a separation of genotypes that seems to be based on heterotic groups: Stiff stalk (B73, B84) as Cd excluder and Lancaster (Mo17, Os6-2) as Cd non-excluder.

## 5. Conclusions

Results of our study showed large genotype variation in Cd accumulated in leaves, selected genotypes separated into two distinct groups by ICP analysis: high accumulating (Os6-2, Mo17) and low accumulating (B73, B84). Accumulated Cd caused changes in plant's photosystem where B73 genotype expressed the smallest changes, while other three genotypes were more affected by Cd, mostly visible as an increase in dissipation energy ($\varphi D_0$, $DI_0/RC$), decreased density of reaction centers (RC/ABS) and decreased contributions of light reactions and electron transport for primary photochemistry ($TR_0/DI_0$ and ($ET_0(TR_0–ET_0)$), respectively) suggesting blockage of electron transport from reaction centers to quinone (re-oxidation problems of primary quinone acceptor ($Q_A$)). Consequently, decreases in performance indexes ($PI_{ABS}$, $PI_{total}$) were observed. From the results of chlorophyll *a* fluorescence measurements and ICP analysis for Cd in ear-leaves, it can be concluded that in the Cd-excluder group, B84 inbred line was sensitive to elevated Cd concentration in soil while inbred line B73 was not. In the Cd-nonexcluder group Cd5 treatment had the greatest effect on photosynthetic performance of Mo17 and OS6-2 genotypes, where Os6-2 line, which accumulated more Cd in ear-leaves, had lower overall performance than Mo17. Our results suggest that there could be at least two possible strategies to utilize these results in maize breeding program. First would be to identify tolerant cultivars to Cd toxicity bred for the Cd contaminated areas (tolerant Cd excluders such as B73). The other would be to identify cultivars capable of high uptake of toxic trace metals without exhibiting toxicity, for remediation of contaminated areas. However, studies with broader germplasm and high-throughput techniques are needed.

**Supplementary Materials:** The following are available online at http://www.mdpi.com/2073-4395/10/7/986/s1, Table S1. Definition of terms and formulae of JIP-test parameters and expressions; Table S2. Results of analysis of variance for JIP-test parameters and Cd and Zn contents.

**Author Contributions:** Conceptualization, D.Š. and M.F.; methodology, D.Š., M.F., V.G.; formal analysis, M.F., V.G. and Z.L.; investigation, M.F.; writing—original draft preparation, M.F.; writing—review and editing, M.F., D.Š. and V.G.; funding acquisition, D.Š. All authors have read and agreed to the published version of the manuscript.

**Funding:** This research was funded by the EU project "Biodiversity and Molecular Plant Breeding", grant number KK.01.1.1.01.0005, of the Centre of Excellence for Biodiversity and Molecular Plant Breeding (CroP-BioDiv), Zagreb, Croatia.

**Conflicts of Interest:** The authors declare no conflict of interest. The funders had no role in the design of the study; in the collection, analyses, or interpretation of data; in the writing of the manuscript, or in the decision to publish the results.

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
