# Peer review of "Genotypic Variability of Photosynthetic Parameters in Maize Ear-Leaves at Different Cadmium Levels in Soil"

_agronomy, doi:10.3390/agronomy10070986_

Round 1

Reviewer 1 Report

Brief summary: 

The manuscript investigate the genotypic variability of photosynthetic parameters in maize ear-leaves in soils diferrentially polluted by the non-essential metal cadmium.  The authors present results of 2 years experiments, detecting effects of Cd on photosynthesis, by analyzing chlorophyll a fluorescence and photosynthetic pigments. Results shown a genotype-dependent response. In general, the manuscript has some potential, but is a bit confusing and not very fluent to read in some points, leading to misinterpretation/misunderstanding (for more details se bulleted point comment below).

Specific Comments:

Comment #1: In the Introduction section, I suggest to authors to report the natural concentration of Cd founded in the polluted soils, as well as the toxicity threshold for higher plants.

Comment #2: In the Introduction section (row 50-52) you only speak about the positive effects of Zn. But in high concetration can also be toxic: pelase add a reference regarding that.

Comment #3: In the Introduction section, I would short the OJIP explanation: it's too long and detailed for being in the introduction section.

Comment #4: Still in the Introduction section, there is a complete lack of information about Cd researches in maize: please add some references.

Comment 5#: In the Materials and Methods section, row 89, add the extended version of WRB before the acronym.

Comment 6#: In the Materials and Methods section, row 98-100, in my opinion there is no needs to explain how you did the solution. Just state the final concentrations.

Comment 7#: At row 105 of the Materials and Methods section, add "plant" before "genotype", to make it more clear.

Comment 8#: From row 105 to row 106 (Materials and Methods), add if you analyze the Cd concentration before or after the xperiment. Written like that is not clear.

Comment 9#: From row 110 to row 111 (Materials and Methods), is not clear how many biological and technical replicates did you use, nor the experimantal design that you setted up. Please clarify that.

Comment 10#: Row 111-112 (Materials and Methods), how did you decide the watering amount? did you measure evapotranspiration or the water use efficiency? otherwise you could also apply a drought stress.

Comment 11#: At row 128 specify why you measured all the Chl a parameters in the upper side of the ear leaves, and not also in other parts of leaves. Please, also here be more cleare regarding the biological and technical replicates: I suggest to authors to use the "n=x" notation in all the manuscript to indicate replicates; it's more clear.

Comment 12#: In the "Photosynthetic pigments and dry weight analysis" paragraph of the Materials and Methods section, add which pigments you analyzed, which standard did you use for the calibration, and the calibration curve.

Comment 13#: at the beginning of the Results section, you said that you have used a multifactorial ANOVA approach. But which ANOVA? two-way, three way? please clarify that. Moreover, all over the manuscript I suggest to add the ANOVA tables with all the interactions (or if you prefer add it in the supplementary materials).

Comment 14#: As a general observation, some results are presented for both years of the experimentation (2012 and 2013), some others are presented only for 1 year, and some are the average of both years. That is not really coherent: please, clarify that.

Comment 14#: The radar plot for photosynthetic parameters (Fig 1) is a good way to present those data. But why didn't you plot significance of the statystical analysis on it? it would reinforce everything (if there is significance).

Comment 15#: At row 299 (Results) you state that "Fig. 3 revealed that accumulation of Cd is genotype dependent". To say that you need to plot the statistical results on the graph, otherwise is difficult to see it.

Comment 16#: At row 399 (Results), you remind to Table 4, but there isn't any Table 4 in the manuscript. Please add it.

Comment 17#: Conclusion are not bad, but I suggest to check again after all the comments provided, especially regarding statistics.

Author Response

Dear reviewer, 

thank you for your time and effort in review of our manuscript.

Please find the attached responses to your questions and concerns (docx file).

Sincerely,

in the name of all co-authors,

Vlatko Galic

Reviewer 2 Report

Interesting work possible to publish, but contains some errors.
Methodology:
- please complete the reagent companies and equipment companies where the analysis was performed
- please complete when soil samples were taken and from what depth
- why the Zn content was measured, it should be developed in the introduction
- why measurements on ear-leaves and not on normal leaves need to be explained
Throughout the text:
1. replace heavy metals with trace metals.
2. correct punctuation errors, e.g. page 6 line 228 and stylistic
3. improve the order of citations that are not in order, e.g. line 457 page 14; line 458 page 14.
4. line 454 page 14 - is there a Cd-pyhtochelatin error?
5. put square brackets line 485 page 15, line 486 page 15
6. citation 42. Maksymiec 1996b should be 1996
7. Errors in citations of literature that do not comply with the magazine's requirements, e.g. items 48, 29

Author Response

(The authors gave the same response as above.)

Reviewer 3 Report

In my opinion, this study is very interesting and it has been carried out correctly. However, some issues should be corrected:

The format of references in the text is not consistent. Besides, the references must appear in the text as numbers: “In the text, reference numbers should be placed in square brackets [ ], and placed before the punctuation”

L42-43. In my opinion, this sentence has no sense in the manuscript because you don’t analyze the role of the developmental stage. I recommend to delete it

L57-58. This sentence is out of context. The paragraph is about photosynthesis but this sentence is about the effect of soil on heavy metal uptake and toxicity. You should delete this sentence or move it to another part of the text

L64 and L70. You repeat “known as JIP test” twice, you should delete one

Use subscript for 3 in CaCO3 in table 1

In Fig. 1, You should make the legend’s colors bigger because it is difficult to see it

L203-204. The results discussed in this sentence are not supported by any table or figure. You may include them even as supplementary material

L206. Table S2 is mentioned here but no Table S2 has been provided. I think that it is a mistake and that it refers to Table S1

L276-277. “B73 line is the only one where dry weight (DW) has decreased with increasing Cd levels”. This sentence is not correct because 3 DW did not decrease with increasing Cd levels in line B7. Please correct it

L281. Use subscript for 2 in CdCl2

In table 2 you use for the first time the separation between: “Cd excluders” and “Cd non-excluders” but no explanation is provided until later in the text. I recommend to put section 3.4 about Cd and Zn as section 3.1 and thus explain the classification “Cd excluders” and “Cd non-excluders”, so that the results can be better understood

Use Zinc abbreviation (Zn) throughout the text

L299. I think that this line refers to Fig. 4 and not to Fig. 3

Fig. 5. Correct “cinc” in the Y axis of 2012 figure

L399. There is no Table 4, correct it

The first paragraph of 4.1. Chlorophyll fluorescence is too long. I recommend to divide it in 2-3 paragraphs

L395. Fig. 5 is not correct. I think that it refers to Fig. 3 or Table 3

L429. I recommend separating the paragraph in two by this line as it speaks of a different parameter

L441-442. Here you mention for the first time “Lancaster” and “Stiff Stalk”. I recommend that you specify that these names refer to the heterotic groups as you do in the lines 473-474. This will improve the reader's understanding

There is a lot of information at the beginning of point 4.3. ICP-OES analysis. You should move this information to the introduction or delete it

I think you did not explain why lower doses of Cd (0.5 and 1) affect more negatively to B73 genotype than the higher dose (5). You should provide a possible explanation for this result in the discussion section

In the discussion there is no information about the possible role of the Zn and Cd transporter HMA4. This information was addressed by Navarro-León et al. (2019) in the article entitle: Possible role of HMA4a TILLING mutants of Brassica rapa in cadmium phytoremediation programs. You could add 1-2 sentences about the possible role of HMA4 transporter in genotypic variability of Zn and Cd accumulation

In the conclusions I recommend you to write 1-2 sentences at the end reflecting the significance of the study, potential applications, or possible future studies.

The formatting of the references is not consistent, please check it

Author Response

(The authors gave the same response as above.)
